# Fire Danger Assessment Using Moderate-Spatial Resolution Satellite Data

**Nataliia Kussul** [1,2] [ID], **Oleh Fedorov** [3], **Bohdan Yailymov** [2], **Liudmyla Pidgorodetska** [3], **Liudmyla Kolos** [3], **Hanna Yailymova** [1,2,*] [ID] and **Andrii Shelestov** [1,2]

1  Department of Mathematical Modelling and Data Analysis, National Technical University of Ukraine 'Igor Sikorsky Kyiv Polytechnic Institute', 03056 Kyiv, Ukraine
2  Department of Space Information Technologies and Systems, Space Research Institute NAS Ukraine & SSA Ukraine, 03187 Kyiv, Ukraine
3  Department of System Research of Space Activity, Space Research Institute NAS Ukraine & SSA Ukraine, 03187 Kyiv, Ukraine
*  Correspondence: yailymova.hanna@lll.kpi.ua

**Abstract:** Fire is one of the most common disturbances in natural ecosystems. The analysis of various sources of information (official and unofficial) about the fires in Ukraine (2019–2020) showed a lack of timely and reliable information. Satellite observation is of crucial importance to provide accurate, reliable, and timely information. This paper aims to modify the index of fire danger of a forest's FWI by increasing its precision, based on the use of higher spatial resolution satellite data. A modification of the FWI method involves the utilization of the soil moisture deficit, in addition to the six subindices of the FWI system. In order to calculate the subindices values, weather data from the Copernicus Atmosphere Monitoring Service were used. Soil moisture deficit is calculated using Sentinel-1 radar satellite data on the water saturation degree of the soil surface layer and geospatial parameters from the 3D Soil Hydraulic Database of Europe. The application of the proposed methodology using the specified satellite, weather, and geospatial data makes it possible to assess fire danger on a continental scale with a spatial resolution of 250 m, 1 km, and a daily temporal resolution. Validation of the proposed method for modifying the FWI system demonstrates an improvement in the precision and relevance of fire danger prediction.

**Keywords:** improved fire danger index; geospatial data; fire danger assessment; Fire Weather Index; satellite data; soil moisture deficit

## 1. Introduction

Global climate change is one of the challenges of our time, leading to an increase in the number of catastrophic natural phenomena and anthropogenic emergencies. Among them, the dangers associated with increased fires in natural ecosystems play an important role. Timely action to prevent and extinguish fires is a pressing challenge in reducing ecological and economic losses. Remote sensing is a generally recognized innovative approach to increasing the effectiveness of forecasting, preventing, and managing risks in this area. In recent years, there have been convincing examples of the successful use of satellite observations due to their repeatability, coverage of vast territories, and the ability to provide accurate, reliable, and timely information.

The fire emergency management examples in Ukraine illustrate the relevance of developing and implementing new methods [1,2]. Emergencies in Ukraine in the autumn of 2019 (burning stubble, peat fires, and adverse atmospheric conditions resulted in a sharp deterioration in the atmospheric air state) and in spring 2020 (very extensive fires in Chernobyl zone forests) show that there are no sources of timely and reliable information about the problems' status at the state administration level. In particular, an analysis of the information about fires in spring 2020 showed that the data of different sources varies



significantly. According to official data, the area of combustion is hundreds of hectares; according to unofficial data, it is thousands of hectares.

There are many studies on fire risk assessment (e.g., [3,4]), and there are many modern information systems for fire danger assessment and fire monitoring using satellite data at a global and regional level: the Canadian Forest Fire Danger Rating System (CFFDRS) [5], the European Forest Fire Information System (EFFIS) [6], NASA's Fire Information for Resource Management System (FIRMS) [7], etc.

The authors of [3] propose a comprehensive fire risk assessment system using geoinformation technology to obtain a spatial assessment of fire risk conditions. Fire danger conditions included human factors, lightning probability, the fuel moisture content of both dead and live fuels, and propagation potential. The sources for the main inputs of the fire risk assessment system were field measurements and remote sensing data. The model was applied at a national level at a 1 km spatial resolution. The information systems of [5–7] allow one to analyze the fire situation in an operational mode on the basis of daily maps of fire danger levels and their forecasts (up to 10 days), using meteorological models, maps of temperature anomalies, and fire perimeters. However, the operational nature of providing informative services by such systems has a number of limitations. To ensure the high temporal resolution of fire situations, satellite data products of low spatial resolutions, mainly MODIS and VIIRS products, are used. Global and regional meteorological models with spatial resolutions in the order of ten kilometers are used to estimate and forecast fire dangers according to meteorological data, which leads to the low spatial resolution of the constructed maps.

Currently, two subsystems of the CFFDRS—the Canadian Forest Fire Weather Index (FWI) System and the Canadian Forest Fire Behavior Prediction (FBP) System—are being used extensively in Canada and internationally [5]. The FBP System provides quantitative estimates of potential head fire spread rate, fuel consumption, and fire intensity, as well as fire descriptions [8]. With the aid of an elliptical fire growth model, the FBP system gives estimates of fire area, perimeter, perimeter growth rate, and fire behavior at the head, flanks, and back of a fire. The data sources for daily maps of estimates of the FBP system are, in particular, data from the FWI system and fuel types. The map of FBP fuel types was derived primarily from forest attribute data [9] based on satellite imagery acquired by NASA's MODIS sensors. Fuel types were assigned based on vegetation type, tree species, crown closure, stand height, and other attributes. This fuels map gives only a general idea of the fuel types present and is not suitable for operational fire management because of the moderate resolution and limited scope of the input data [8]. The Index FWI is also used in EFFIS [6] and the Copernicus Emergency Management System (CEMS) [10].

Despite its relative universality and worldwide distribution, the FWI has significant limitations due to the fact that its components are defined only by meteorological data and not by fuel type or, at least, the type of land cover. Therefore, the urgent problems of increasing the efficiency of fire danger assessments are both the increase in spatial resolution and the precision of assessments.

The purpose of this research is to modify the index of fire danger of a forest's FWI by increasing its precision, based on the use of higher spatial resolution satellite data and an index generalization to other types of natural ecosystems. The idea behind the improvement of the FWI methodology is to involve a factor of moisture deficit in the soil in the formation of a generalized fire danger index.

## 2. Methodology

### 2.1. Basic Approaches to Fire Danger Assessment

The method presented in this paper is based on the Fire Weather Index (FWI) [11]. The system of FWI calculation was developed in the 1970s, after which it was modified and improved several times [11]. The system consists of six standard components—three fuel moisture codes (FFMC, DMC, DC) and three fire behavior indices (intermediate fire behavior components ISI and BUI and the general index FWI). These six components are

weather based and provide numeric assessment of the relative probability for wildland fire. The source data for the sub-components' calculation are noon temperature, relative humidity, rainfall (accumulated in 24 h), and/or wind speed.

The FWI system and its components have proved to be suitable for reflecting various aspects of fire activity, in particular, the FFMC correlates well with fire occurrence, the ISI with burnt area, and the BUI and FWI with fire activity in general [12].

The three fuel moisture codes follow daily changes in the moisture content of three classes of forest fuel with different drying rates. Each moisture code is calculated in two phases—one for wetting by rain and one for drying—and is arranged so that higher values represent lower moisture contents, hence greater flammability [12]. Consider them in some more detail.

The Fine Fuel Moisture Code (FFMC) is a numeric rating of the moisture content of litter and other cured fine fuels and it is an indicator of the relative ease of ignition and flammability of fine fuels. It is calculated by changing its previous value by an amount that represents any significant precipitations or atmospheric moisture/drying that took place during the day.

The Duff Moisture Code (DMC) is a numeric moisture indicator in the weakly compacted organic layer (duff), which is spread, moderate depth, and is an indicator of fuel content in a duff layer and in medium-sized wood material. The calculation of DMC is almost the same as the calculation of FFMC—first the duff layer moisture content of the previous day is calculated, then the corrections are applied taking into account the rainfall phase or drying phase of the day and the effective day length [12,13].

The Drought Code (DC) is a numerical indicator of moisture in a deep dense organic layer (deeper layer represented by the DMC) and is an indicator of seasonal influence of drought on forest fuel and smoldness in deep duff layers and large logs. The code is defined for the current day by its value of the previous day and the potential evapotranspiration of the deep duff layer (depending on the rainfall or drying phase), which is a function of temperature and a seasonal day length adjustment.

The FFMC, DMC, and DC are supposed to be calculated on a daily basis. The meteorological data used for its calculation have to be recorded at noon (for fire danger prediction at about 4 pm). The three codes calculation starts in regions normally covered by snow in winter on the third day after snow has essentially left the area. In regions where snow cover is not a significant feature, the calculation starts on the third successive day with a noon temperature greater than 12 °C [12,13].

The second level of the FWI system consists of three fire behavior indices [12]. Two intermediate fire behavior indexes represent fire spread rate and amount of available fuel.

The Initial Spread Index (ISI) is a numeric indicator of the expected rate of fire spread, when the fine fuel (litter) is dry but drying in depth is not advanced. The ISI depends on the wind speed (limited to a maximum of 100 km/h) and the FFMC. Like other components of the FWI system, the ISI does not take into account the fuel type.

The Buildup Index (BUI) is a numeric indicator of the total amount of fuel available for combustion, which combines the DMC and DC. The BUI is generally less than twice the DMC value, and moisture in the DMC layer is expected to help prevent burning in the material deeper down in the available fuel. The DMC reacts to rainfall or a lack of rainfall quicker than the DC, which represents the deeper duff layer [12,13].

The final fire behavior index, the Fire Weather Index (FWI), combines the ISI and BUI and it is numeric indicator of fire intensity. The FWI is a positive number, with low values indicating low fire danger weather conditions and higher values indicating higher fire danger weather conditions.

These six standard components of the FWI System are predictors of daily fire potential. Because one value per day is determined for each component, the FWI System does not indicate hour-by-hour changes, nor does it account for variations in fuel type from season to season or from place to place [12]. However, it does provide reference scales that permit comparisons of fire danger with other days and other locations. The FWI System makes

it possible to reconstruct past fire danger conditions if suitable historical weather records are available.

The method of calculation of the components of the FWI system is rather cumbersome and is detailed in [13–15]. The components of the FWI system appear in 6 fire danger classes: very low, low, medium, high, very high, and extreme.

### 2.2. Methodology for Modifying the Fire Danger Index FWI

It is proposed to use a moisture deficit in the soil to improve the FWI methodology. This is specifically the SMD (Soil Moisture Deficit) [16] parameter, which depends on soil type and can be determined from remote data in the microwave range. Involvement of SMD to the established set of parameters of the FWI system (schematically represented in Figure 1) resolves 2 problems. First, the spatial resolution of the initial information (facilitated by the use of the relevant geoinformation data), and thus the localization of the predictive estimate is significantly increased. Second, the relevance of the final assessment is increasing, as the established set of sub-indicators is valid for the forest area and the SMD is informative for a surface that does not have high vegetation. Such an approach seems essential for the territory of Ukraine, where significant forest, steppe, and forest–steppe areas are bordered.

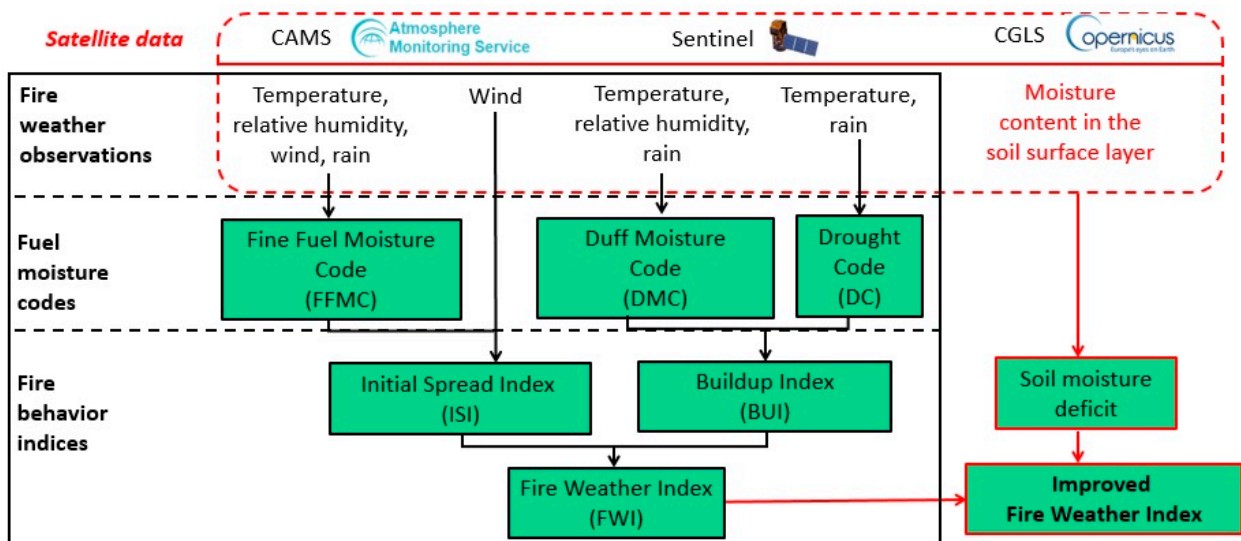

**Figure 1.** Modification of the Fire Weather Index FWI using satellite data.

The implementation of the above procedure in such an approach requires the assimilation of evaluation data on the basis of five components according to the established methodology and components based on SMD data. To determine the last component, it is also necessary to develop a procedure for calculating the generally accepted index of moisture deficit in soil using satellite data. Thus, the proposed procedure for improving the FWI index includes saving the general scheme of its definition and entering the procedure of taking into account the established parameters on the basis of remote data. SMD values, expressed in physical units, are converted into scores, as is the value of the FWI index. As a result of folding these 2 criteria with certain weights into a single criterion, the evaluation of the generalized index of fire danger in points is obtained.

A detailed description of the determination of the surface soil moisture deficit and of the improved fire danger index can be found in the Supplementary Materials section.

*2.3. Application of Satellite and Geoinformation Data as Input to the Calculation of a Fire Danger Index*

In order to produce fire danger maps based on the modified index $FWI_{impr}$, it is necessary to collect large amount of satellite data, to carry out their correct processing using deep learning methods and cloud processing of information, to create a ground data set for validation, and to verify the results [17].

First of all, in order to calculate the values of the standard index $FWI$ for the current day, it is necessary to download daily data for the previous period. In particular, data such as air temperature at the height of 2 m, wind speed at the height of 10 m, and relative humidity of air and precipitation quantity for the last 24 h. These weather data are downloaded free of charge using the Copernicus Atmosphere Monitoring Service (CAMS) [18].

Secondly, to calculate the moisture deficit of the upper layer, geospatial data are needed, in particular, regarding the physical and hydraulic characteristics of soils, as well as radar satellite data that are relevant for the day for which the fire danger is assessed.

The volumetric field capacity required to calculate the water content at the field capacity and directly the moisture deficit in the soil surface layer was obtained from the 3D Soil Hydraulic Database [19] presented by the European Soil Data Center ESDAC in 2017 [20,21]. The database contains the geospatial data of the soil hydraulic properties at 7 standard depths from 0 to 2 m with both a 1 km and a 250 m spatial resolution for the territory of Europe.

The global data set of retrospective analysis MERRA-2, created for the study of physical processes in the Earth's climatic system, was used as data on soil porosity [22].

The soil saturation degree data required to calculate the actual water content in the surface soil layer according to Equation (S7) can be obtained from Surface Soil Moisture (SSM) on the Copernicus Global Land Service (CGLS), based on satellite radar data from Sentinel-1 [23].

The testing and validation of the proposed approach to improving the index were carried out on the territory of Ukraine, based on satellite observations for the period of 10–15 August 2021 [24].

For the calculation of the standard fire danger index $FWI$, CAMS data from 15 August 2021 were used regarding air temperature at a height of 2 m, wind speed at a height of 10 m, the relative humidity of the air, and the precipitation quantity for the previous 24 h.

To calculate the moisture deficit in the surface soil layer based on satellite data, the SSM CGLS product was selected for the period of 10–15 August 2021 with a spatial resolution of 1 km and the data from the database of hydrological properties of soils of 250 m [21] and a set of global data [22] were used.

To calculate the improved fire danger index $FWI_{impr}$, taking into account both meteorological data and data on soil moisture deficit, depending on its type, a linear convolution of partial criteria of fire danger was formed according to Equation (S17). Preliminarily, the coefficients k1 and k2 were taken equal to 0.5, thus giving an equal value to both the meteorological factors of the standard index FWI and the dryness factor of the soil surface layer, depending on its type.

The developed fire danger assessment method, based on the improved index $FWI_{impr}$ and the methodology described above, was implemented in the Google Earth Engine cloud platform.

To compare the results of calculating the fire danger levels using the proposed method, the product FWI of the Copernicus Emergency Management System (CEMS) [10], which is freely available, was selected. This product was calculated using a standard algorithm and was based solely on weather data. The spatial resolution of the product was 0.25 geographic degrees.

The conformity assessment of the $FWI_{impr}$ fire danger level map to the FWI (CEMS) fire danger level map was carried out on the basis of the confusion matrix, which was a cross-tabulation tool [25].

Additionally, a comparison was made on the relationship between the number of fires in ecosystems and in open areas that occurred on 15 August 2021, according to operational data per day from the regional departments of the State Service for Emergency Situations of Ukraine (SES of Ukraine) [26], with the results of the fire danger assessment by FWI (CEMS) and $FWI_{impr}$. Linear regression was used for this purpose, and the coefficient of determination and residual standard error were used to evaluate the relationship's goodness of fit. The number of fires was the dependent variable, $FWI_{impr}$ was the independent variable. Data of the SES of Ukraine were the statistical data on the number of fires in a certain region per day and the places where the fire occurred (near which settlement and the area of the fire, without indication of the certain point of the fire).

## 3. Results and Discussion

The result of the FWI index calculation, based solely on weather data, is presented in Figure 2.

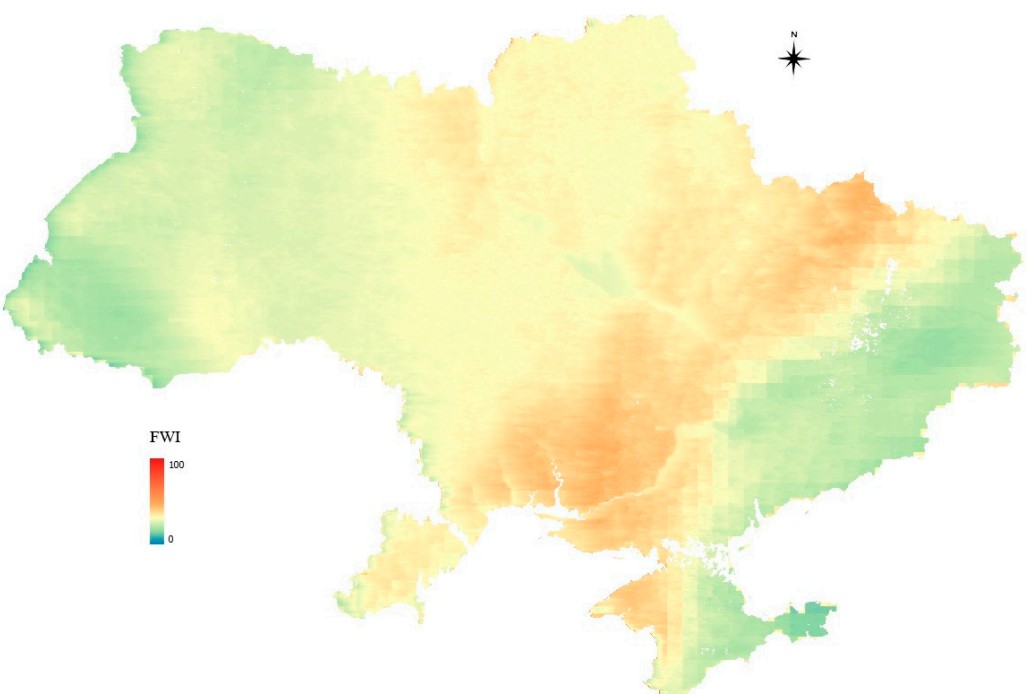

**Figure 2.** FWI index as of 15 August 2021 (spatial resolution 1 km).

The result of calculating the moisture deficit in the soil surface layer up to 5 cm deep (in mm) was converted to a 100-point scale, and it is shown in Figure 3.

The result of calculating the improved fire danger index $FWI_{impr}$ is shown in Figure 4.

As can be seen from Figure 4, the result is more detailed than the standard index FWI (Figure 2) and takes into account equally the weather data as well as the level of moisture deficit in soil depending on its type. The improved fire danger index $FWI_{impr}$ is a rating of fire danger intensity from 0 to 100 and is classified into six levels, the thresholds of which are also defined as the arithmetic mean of the thresholds of the standard index FWI and moisture deficit in the soil surface layer, normalized to a 100-point scale.

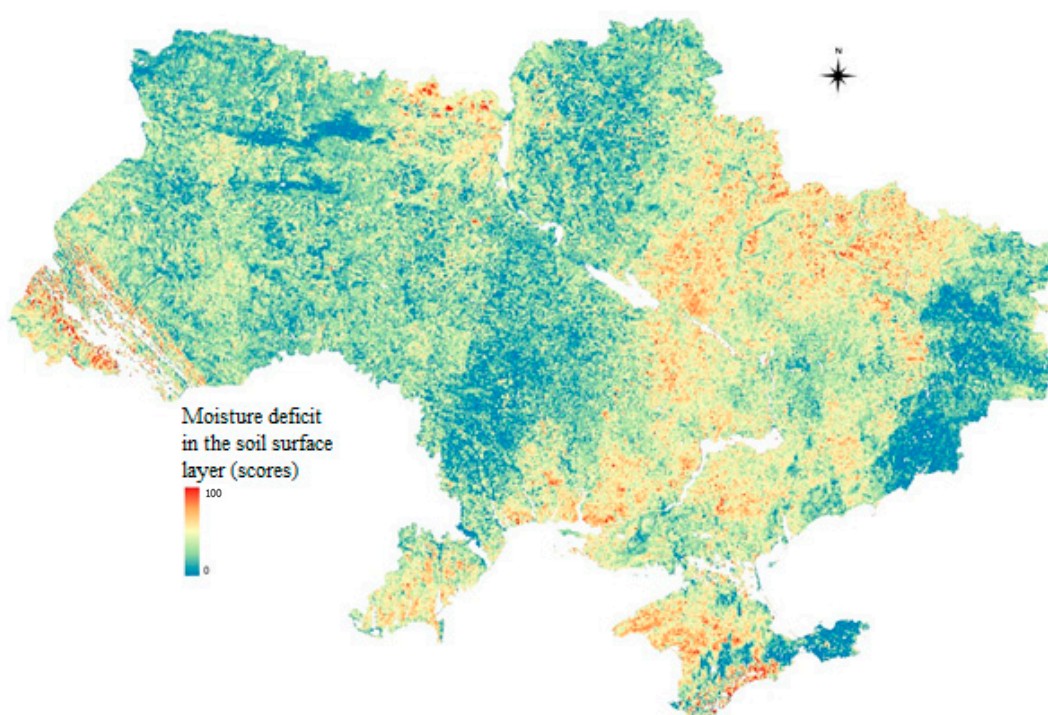

**Figure 3.** Moisture deficit in the soil surface layer (scores), 10–15 August 2021 (spatial resolution 250 m).

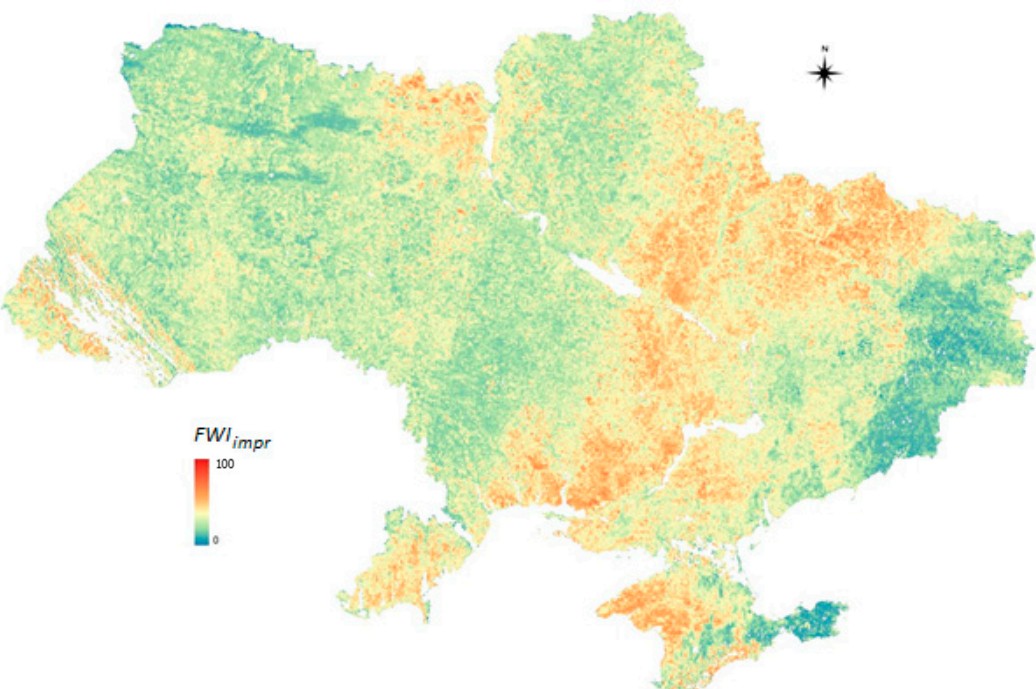

**Figure 4.** Improved index $FWI_{impr}$ on 15 August 2021.

The classification of the improved index raster into six levels (Table S2) of fire danger was carried out using threshold segmentation and is shown in Figure 5. Class 1 corresponds to a very low level of fire danger (green), class 2—low level (yellow), class 3—moderate level (orange), class 4—high level (red), class 5—very high level (purple), and class 6—extreme fire danger (black).

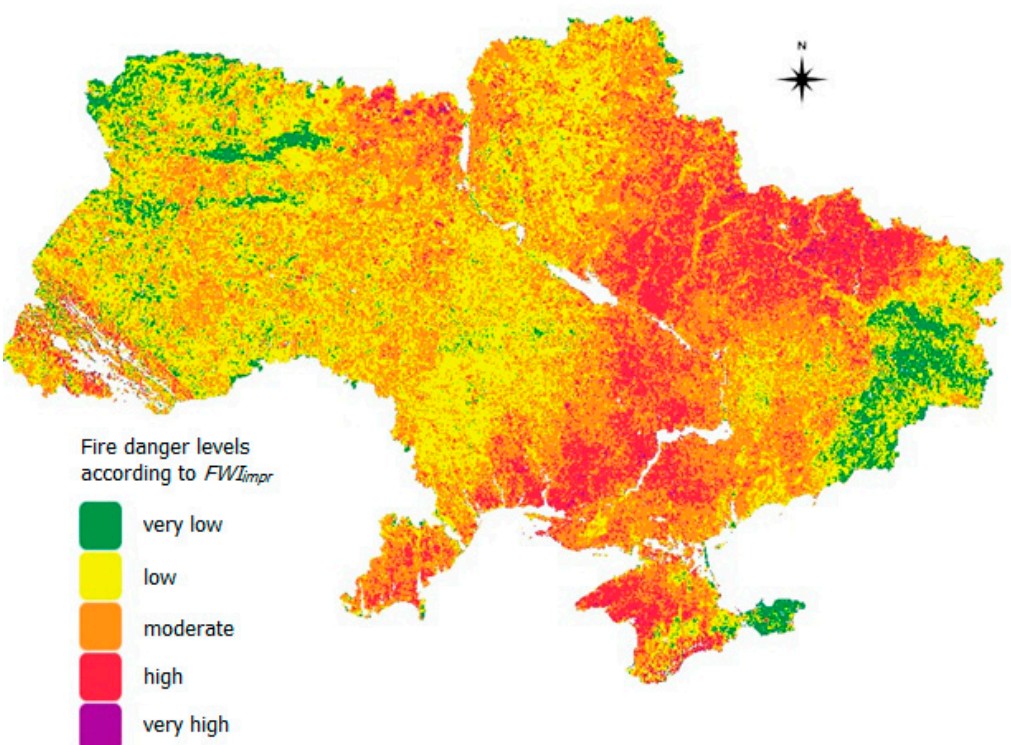

**Figure 5.** Map of fire danger levels according to the improved fire danger index $FWI_{impr}$ on 15 August 2021.

As can be seen, the applied calculation procedure resulted in a map of five fire danger levels according to index $FWI_{impr}$—from very low to very high. There is no extreme fire danger class on the territory of Ukraine on 15 August 2021.

The map of fire danger levels, calculated according to FWI (CEMS) data [10] for the territory of Ukraine on 15 August 2021, is shown in Figure 6.

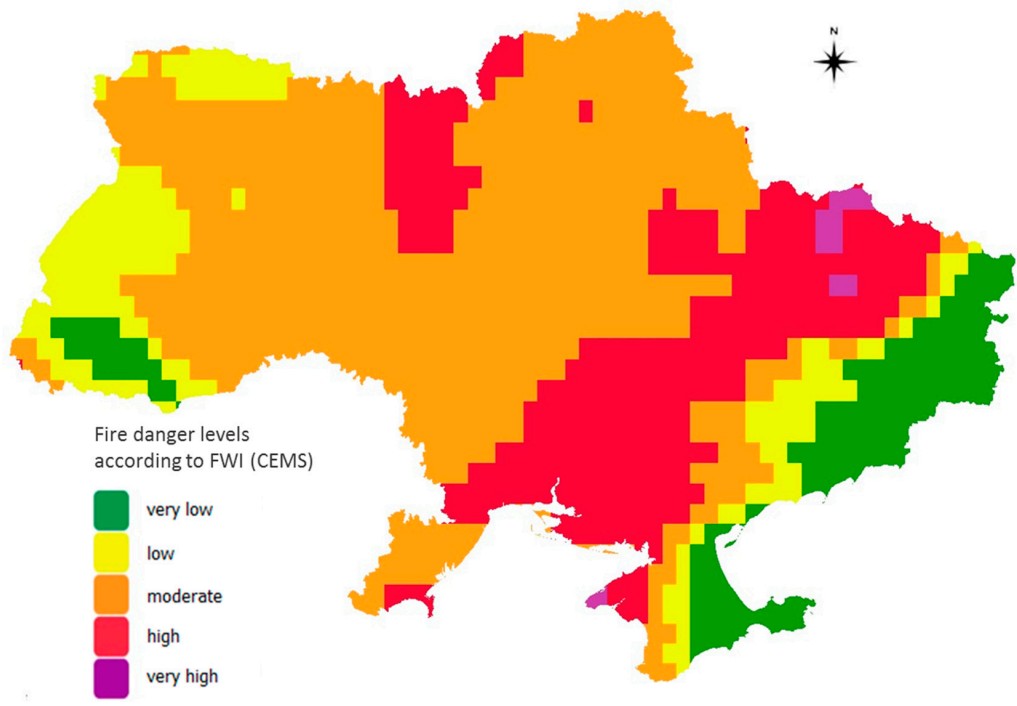

**Figure 6.** Map of fire danger levels for the territory of Ukraine according to FWI (CEMS) on 15 August 2021.

Conformity of the fire danger level map according to the $FWI_{impr}$ with the FWI (CEMS) fire danger level map based on the confusion matrix was 42.23%. This percentage of compliance is explained both by a significant difference in the spatial resolution of the compared maps (0.25 geographical degrees and 250 m, respectively), and by the difference in the data sources on the basis of which they are built—exclusively weather data for the standard index FWI (CEMS) and a combination of weather, satellite data, and geospatial information on the hydrological properties of the surface layer of the soil for an improved index $FWI_{impr}$.

Comparison of the dependences of the number of fires in ecosystems and in open areas by regions on the values of the $FWI_{impr}$ and the FWI (CEMS) on 15 August 2021 showed (Figure 7) that the proposed $FWI_{impr}$ index correlated better with the number of real fires by the regions compared to FWI (CEMS). The Pearson's correlation coefficients were R = 0.81 and R = 0.78, the residual standard errors were 4.95 and 5.25, and *p*-values were 0.0000129 and 0.0000382, respectively.

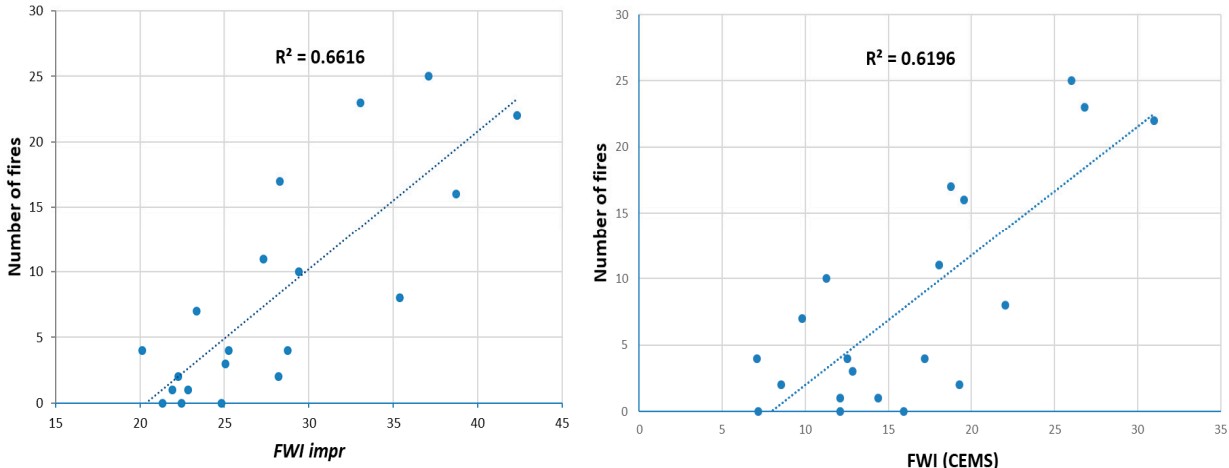

**Figure 7.** Relationship between the number of fires according to statistics by regions of Ukraine and, respectively indices $FWI_{impr}$ and FWI (CEMS) on 15 August 2021.

Operational data on the number of fires on 15 August 2021 were obtained from the 20 regional departments of the SES of Ukraine (Figure 8). Shaded areas in Figure 8 are the regions with no data about the number of fires in ecosystems and open areas.

Additionally, using data from the SES of Ukraine on fires by regions for the period June–September 2021 [26], the results of assessing the fire danger by the index were analyzed (Figure 9).

The assessment of the dependence of the number of fires on the value of the index $FWI_{impr}$ (for the summer period, the Pearson's correlation coefficient R = 0.65, for the autumn R = 0.7) shows that this index correlates well with real fires in the regions observed in different periods of the year. The validation carried out is of the nature of a preliminary study, since more relevant data should be included for a correct comparison. First, a certain number of fires includes all facts without exception (in particular domestic and others, unrelated to the estimated index). Second, a more detailed time series, rather than monthly averages, should be used for the correct comparison. Considering these circumstances, the given comparison estimates look quite encouraging regarding the further use of the applied approach.

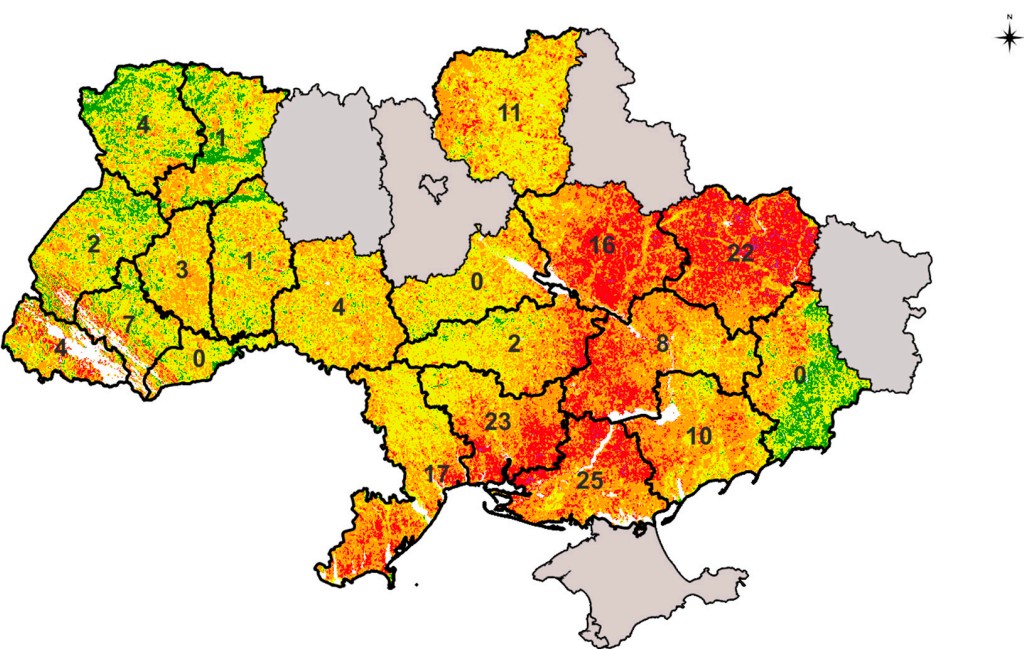

**Figure 8.** Overlaid regions of Ukraine on the map of improved fire danger index $FWI_{impr}$ on 15 August 2021 (the numbers on the map represent the number of fires in ecosystems and open areas per regions according to data of the SES of Ukraine).

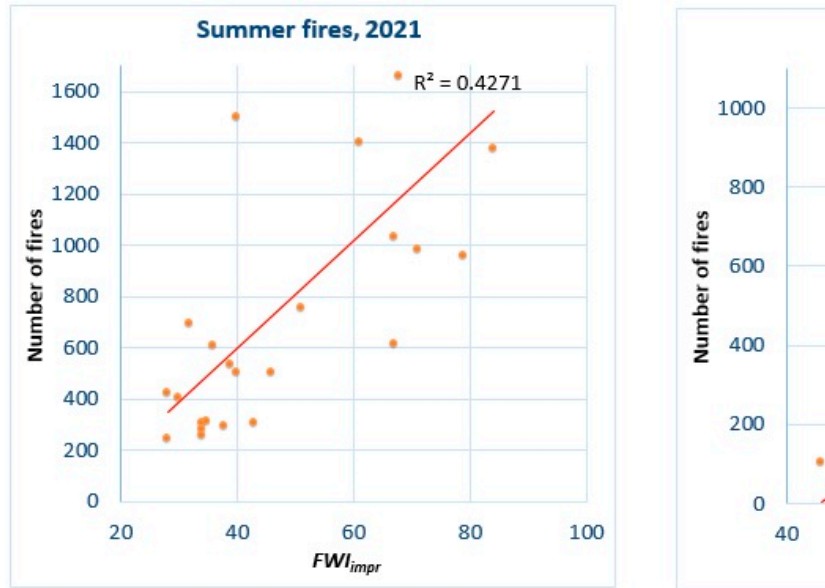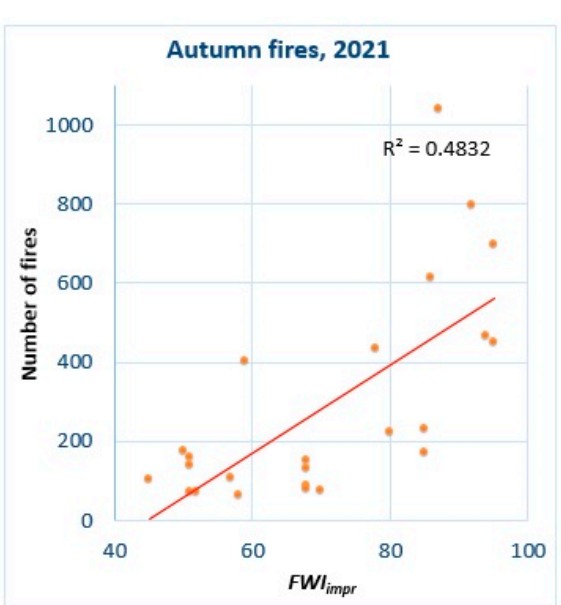

**Figure 9.** Relationship between $FWI_{impr}$ and the number of fires by region of Ukraine.

Therefore, the developed index $FWI_{impr}$ can be used to support decision-making in the SES of Ukraine and other government agencies in order to prepare in advance for possible adverse situations [27].

The main limitations of the proposed approach are due to limitations of the SSM from the Sentinel-1 CSAR [23]. The product algorithm allows deviations biases in the soil moisture during the vegetation full development period over areas with high density (e.g., forests). Soil moisture cannot be retrieved over deserts and tropical forests. While a terrain correction is performed, it does not completely remove the influence of topography. This limitation comes into effect especially over high mountain ranges, and thus the SlopeMask is applied. In addition, no reliable soil moisture measurements can be done during frozen

or snow-covered conditions. Additionally, the limitation of the proposed approach is the possibility of its implementation for a relatively small depth of the surface soil layer of 5 cm. However, the moisture deficit determined for this depth significantly affects the potential fire danger.

Further refinement of the method will concern the introduction of machine learning to improve the quality of the generalized index as a linear convolution of partial criteria of fire danger.

## 4. Conclusions

1. A method of fire danger assessment using an improved index FWI (Fire Weather Index) is proposed. A modification of the FWI method involves the utilization of the indicator of soil moisture deficit, in addition to the established six components (subindices) of the FWI system, which are predictors of daily potential fire. Soil moisture deficit (SMD) is obtained from the Sentinel-1 radar satellite data concerning the degree of water saturation of the soil surface layer, which significantly affects the possibility of occurrence and course of fires in the natural ecosystem.

2. To calculate the moisture deficit of the upper layer, in addition to satellite data, geospatial data are also used, in particular, regarding the physical and hydrological characteristics of soils. Calculations of the steady-state sub-indices of the FWI system included weather data from the CAMS service. The application of the proposed methodology using the specified satellite, weather, and geospatial data makes it possible to assess the fire danger on a continental scale with a spatial resolution of 250 m, 1 km, and a daily temporal resolution.

3. A scale of fire danger levels is proposed, taking into account the ranges of values of the normalized moisture deficit in the soil surface layer. The scale is ideologically close to the Keetch–Byram drought index scale but differs in particular by the introduction of a level, which corresponds to the negative value of the index.

4. Validation of the proposed method for modifying the FWI system demonstrates an improvement in the precision and relevance of fire danger prediction. It is planned to use the developed method in the information technology being created for assessing fire danger and monitoring fires using satellite data on the territory of Ukraine.

**Supplementary Materials:** The following supporting information can be downloaded at: https://www.mdpi.com/article/10.3390/fire6020072/s1, Table S1: Fire danger levels according to the Keetch–Byram index; Table S2: Fire danger levels according to the normalized surface soil moisture deficit values ranges. References [28–40] are cited in the supplementary materials.

**Author Contributions:** Conceptualization, N.K. and O.F.; methodology, N.K., O.F., B.Y., L.P., L.K., H.Y. and A.S.; software, B.Y., L.P. and H.Y.; validation, B.Y., L.P. and H.Y.; writing—original draft preparation, O.F. and L.K.; writing—review and editing, O.F., L.P., L.K. and H.Y.; supervision, N.K. and O.F.; project administration, N.K. All authors have read and agreed to the published version of the manuscript.

**Funding:** This research was carried out within the framework of the project "Information technology for fire danger assessment and fire monitoring in natural ecosystems based on satellite data" within the competition of the National Research Foundation of Ukraine "Science for the safety of human and society" (2020.01/0268), the project "Deep learning methods and models for applied problems of satellite monitoring" (2020.02/0292) within the competition of the National Research Foundation of Ukraine "Support research of leading and young scientists" from the State budget, and the project and the target program of the National Academy of Sciences of Ukraine "Aerospace observation of the environment in the interests of sustainable development and security"—"Evaluation of indicators of sustainable development for monitoring natural resources based on satellite data" (ERA-PLANET/UA).

**Institutional Review Board Statement:** Not applicable.

**Informed Consent Statement:** Not applicable.

**Data Availability Statement:** Not applicable.

**Conflicts of Interest:** The authors declare no conflict of interest.

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
