# Peer review of "Fire Danger Assessment Using Moderate-Spatial Resolution Satellite Data"

_fire, doi:10.3390/fire6020072_

Round 1
Reviewer 1 Report
Wildfire danger risk assesment is today a quite hot topic and lot of researchers are currently dealing with them. The paper presents one of approaches where wildfire danger risk is calculated based on satelite monitoring. A well known Fire Weather Index is modified and improved using the soil moisture deficit. Authors quite clearly describe their approach and explain why this imnprovement has been important. The paper is well structured and everything is well described.
The only mnissing point is the lack of comparison of this approach for wildfire danger risk assesment based entirely on metheorological data with other approaches for wildfire danger risk estimation that take into account other important factors (see Chuvieco, E., Aguado, I., Jurdao, S., Pettinari, M.L., Yebra, M., Salas, J., Hantson, S., de la Riva, J., Ibarra, P., Rodrigues, M., Echeverría, M., Azqueta, D., Román, M.V., Bastarrika, A., Martínez, S., Recondo, C., Zapico, E., & Martínez-Vega, F.J. (2014). Integrating geospatial information into fire risk assessment. International Journal of Wildland Fire, 23, 606–619).
Therefore Introduction part needs a minor revision where these other, more complex approaches are mentioned and differences between them and meteorolgicaly based approaches are emphasised.
Author Response
Thank you for your review of our manuscript. We have taken note of your comment (lines 49, 54–59 of the current manuscript). Note that we reformatted the manuscript (introduction, methodology, results and discussion, conclusions).
Reviewer 2 Report
This paper describes an attempt to modify the Canadian Fire Weather Index (FWI) to increase its accuracy by including satellite data derived at higher spatial resolution, and to put this modified system to use in Ukraine.The paper describes a potentially interesting attempt to achieve a higher spatial resolution of the used meterological indixes by complementing with remote sensing derived data.
However, the authors seem to lack the required familiarity with the Canadian Forest Fire Danger Rating System (CFFDRS) and the connection between the Fire Danger Rating System and the Fire Behavior Prediction System. So while it is true when the authors state that CFFDRS is “defined only by meteorological data and not by fuel type or, at least the type of land cover” (l 137), they ignore the fact that the second part of the system, the behavior prediction system, is based on fuels, and the weather components of the CFFDRS feed into the calibrated fuel types of the behavior prediction system. Also, when they state that there are no hourly values available for FWI or FDRS - that is not true as there are ways to calculate hourly values, just that these have not been implemented in the Copernicus version of the CFFDRS downloaded by the authors, but it is possible to calculate these with published Open Source Software (https://www.canadawildfire.org/cffdrs-r-package).
After introducing the CFFDRS system, the authors propose to introduce an improvement to the Fire Weather Index (FWI) component of the CFFDRS by adding a moisture deficit term to it. The Fire Weather Index is a compound index that is calculated from its sub-indices which contain information on the (empirically estimated) moisture state of various fuel components, but also wind speed. In the reviewers opinion it does not make sense to add the moisture deficit on top of such a compound index that already inherently contains information closely related to soil moisture. It would make a lot more sense to introduce the additional soil moisture information to one of the sub-indices which are sensitive to soil moisture such as the Drought Code or the Duff Moisture Code.
After introducing the idea of an “improved” FWI, the authors go on to explain “Surface soil moisture deficit determination using the satellite data”, and present a chapter that is crammed with formulas on soil moisture (most of which can be found in common textbooks on the topic) but does say little to nothing on how soil moisture deficit is actually derived from Sentinel 1 radar data.
After that they present the formular on the improved FWI which consists in just adding a soil moisture deficit term (m) to FWI and multiplying both FWI and k with factors k1, and k2 “which can be derived like this “Different methods can be used to determine the coefficients k1 and k2, both using machine learning and without it” (l 249). They go on describing a principal component based approach to derive these weight coefficients, and scale them with another meteo-data derived index, the Keetch-Byram Drought index (which by the way has nothing to do with CFFDRS and is also an empirical index). This index is again introduced at great length with a lot of formulas (also available in textbooks), however, the rationale of the whole exercise is poorly explained and not fully clear to the reviewer.
The actual calculation steps of the improved FWI are then presented in just a few lines (344-367), where finally the data sources used are revealed very briefly, and the reader learns that the Sentinel 1 data processing is actually not done by the authors, instead the Copernicus Global Land Service (CGLS) soil surface moisture product is used.
The next chapter shows the results for a period of 5 days in August 2021. The data of the improved FWI are shown to be more detailed, which is not surprising given the higher spatial resolution of inputs. Data are also compared to a downloaded dataset at 0.25° resolution from Copernicus Emergency Management System (CEMS). Here the authors provide a confusion matrix based conformity analysis between the two systems and report an “overall accuracy” of 42%. The reviewer thinks that this does not make much sense – an accuracy assessment is useful if you have a reference database that is “better” than your product – this can’t be the case if the product to be improved is compared to the “improved” product, and both at very different spatial resolutions – it is hard to see what the actual information value of this comparison is, apart from showing that the results are different. To prove that the new data are indeed better, the results are then compared to reported fires from the regions of Ukraine, and the number of fires are correlated to the recorded FWI and improved FWI classes those fire points fall in (supposedly, no detail is given on the reference data source as to completeness, spatial resolution a.s.o.). It can be debated, whether fire counts are in general a good metric to evaluate FWI performance, though is also done in other studies that take the Canadian FWI to other areas of the world, and at least it can give a first impression of quality. As can be seen in the charts (Fig 8, the charts y axis is labeled in Cyrillic script instead of English), however, there is at best a marginal and probably not significant difference between the coarse resolution FWI product and the improved FWI although the authors claim that this proves that the improved FWI is better. Two further charts are given (in strangely bad print quality) showing the relation for summer and fall fire counts to the improved FWI, but – strangely - no comparison is given to original FWI. Also, the reviewer thinks that for both comparisons, it would be more logical and informative to compare the 1 km FW presented in figure three to the improved product and not the much lower resolution CEMS product.
Taken together these flaws make the paper unfit for publication in its present form. A thorough reworking is required in which the authors should, among other things:
- Demonstrate increased familiarity with the purpose and structure of the CFFDRS and its connections to fire behavior, as well as current work on the issue.
- Reconsider the way the soil moisture deficit estimates are attached to CFFDRS, e.g. meaningful and provable improvement of a sub-index.
- Show an increased proficiency with the datasets they are using to produce their products – e.g. the user guide of the Copernicus soil moisture product states the limitations of the product over forests, which are not treated in the manuscript but have important consequences.
- Elaborate a concise manuscript which details what the authors actually did and show a critical evaluation of their findings and e.g. of intermediate products. The current manuscript is poorly structured and confusing to read, which makes a more in depth evaluation difficult.
- Improve English language, in its present form the manuscript would require extensive English language editing.
- Provide the charts in a correct way (English labels, no blurry charts).
Author Response
Thank you for your review of our manuscript. Authors made changes and corrections to the text of the article, trying to respond to all the comments of the reviewer:
– made some changes demonstrating our familiarity with CFFDRS (as we see it);
– changed the structure of the text and tried to describe the intermediate stages more fully;
– clarified the limitations of the proposed approach in accordance used SSM CGLS product;
– have added some arguments about our approach.
